

# Regional Contributions to Particulate Matter Concentration in the Seoul Metropolitan Area, Korea: Seasonal Variation and Sensitivity to Meteorology and Emissions Inventory

Eunhye Kim[1], Changhan Bae[1], Hyun Cheol Kim[2, 3], Jeong Hoon Cho[4], Byeong-Uk Kim[5], and Soontae Kim[1,*]

[1]Department of Environmental and Safety Engineering, Ajou University, Suwon, 16499, Korea
[2]Air Resources Laboratory, National Oceanic and Atmospheric Administration, College Park, MD, 20740, USA
[3]Cooperative Institute for Climate and Satellites, University of Maryland, College Park, MD, 20740, USA
[4]Environmental Meteorology Research Division, National Institute of Meteorological Sciences, Jeju, 63568, Korea
[5]Georgia Environmental Protection Division, Atlanta, GA, 30354, USA

*Correspondence to*: Soontae Kim (soontaekim@ajou.ac.kr)

**Abstract.** The impact of regional emissions (e.g., domestic and international) on surface particulate matter (PM) concentrations in the Seoul Metropolitan Area (SMA), South Korea and its sensitivities to meteorology and emissions inventories are quantitatively estimated for 2014 using regional air quality modeling systems. Located on the downwind side of strong sources of anthropogenic emissions, South Korea bears the full impact of the regional transport of pollutants and their precursors. However, the impact of foreign emission sources have not yet been fully documented. We utilized two regional air quality simulation systems: (1) a Weather Research and Forecasting and Community Multi-Scale Air Quality (CMAQ) system; and (2) a United Kingdom Met Office Unified Model and CMAQ system. The following combinations of emission inventories are used: the Intercontinental Chemical Transport Experiment-Phase B, Inter-comparison Study for Asia 2010, and the National Institute of Environment Research Clean Air Policy Support System. Partial contributions of domestic and foreign emissions are estimated using a brute force approach, adjusting South Korean emissions to 50 %. Results show that foreign emissions contributed ~65 % of SMA surface PM concentration in 2014. Estimated contributions display clear seasonal variation, with foreign emissions having a higher impact during the cold season (Fall to Spring), reaching ~80 % in March, and making lower contributions in the summer, ~40 % in July. We also found that simulated surface PM concentration is sensitive to meteorology, but estimated contributions are mostly robust. Regional contributions are also found to be sensitive to the choice of emissions inventories.

## 1 Introduction

Regional air quality in East Asia has been a serious concern accompanying this region's rapid economic growth. In recent years, the rapid increase of industrialization and energy consumption in China, especially, has sharply increased the release of anthropogenic pollutants and their precursors (Ohara et al., 2007; Richter et al., 2005; Streets, 2003; Zhang et al., 2009).



East Asia is now one of the most dominant sources of aerosols and trace gases. Most of all, airborne particulate matter (PM) has drawn great public interest due to its adverse effects on human health (Pope and Dockery, 2006), visibility (Watson, 2002; Zhang et al., 2012), and global climate (Wang et al., 2014).

Korea and Japan, located on the downwind side of China, have experienced the impact of transported pollutants and
precursors from that neighboring country, one of the most dominant sources of anthropogenic emissions. Long-range transport in East Asia has been studied to understand the pathways and impacts of dust storms, known as Asian Dust or Yellow Sand (Chun et al., 2001; Chung, 1992; Iwasaka et al., 1983; Kim et al., 2008; Kwon et al., 2002; Sun et al., 2013), and acidic deposition processes (Larssen et al., 1999; Park et al., 2005). Recently, studies have been conducted on the chemical processes underlying PM formation, including the secondary formation of PM components and the transport
patterns related to synoptic meteorology (Chang et al., 2010). However, gaining full understanding of the behavior and impact of PM is challenging, mostly because aerosol formation and interaction are intricate and complex.

Quantitative estimation of the partial contributions made by domestic and international emission sources is very important to maximize the efficiency of emission regulation policy. Several studies have worked to quantify the impact of Chinese anthropogenic emission on Korean air quality. Contributions from the transport of natural and anthropogenic emission
sources have been analyzed using Lagrangian trajectory models and Eulerian chemistry-transport models. Using a back-trajectory-based potential source contribution function, Han et al. (2011) identified the major industrial areas of Eastern China as possible source areas for high $PM_{2.5}$ concentrations in rural sites in Korea. Soil, combustion processes, non-metal manufacture, and secondary $PM_{2.5}$ sources were identified as accounting for 77 % of the total explained variance. Using highly time-resolved measurements, Jeong et al. (2013) also estimated the contributions made by long-range-transported
aerosol in East Asia to carbonaceous aerosol and PM concentrations in Seoul, Korea, concluding that highly industrialized areas of Northeast China (e.g., Harbin and Changchun) and Eastern China (e.g., the Pearl River Delta, Yangtze River Delta, and Beijing-Tianjin regions) are possible source regions for high organic carbon, elemental carbon, and $PM_{2.5}$ in Seoul. Choi et al. (2013) analyzed the chemical composition and possible source of $PM_{2.5}$ collected from coastal areas of Korea using a positive matrix factorization model, demonstrating that secondary particles are dominant in the composition of $PM_{2.5}$, likely
contributed by the major industrial areas in China.

Contributions of Chinese emissions have also been estimated using Eulerian approaches. Lin et al. (2008) established region-to-region source–receptor relationships for sulfur and reactive nitrogen deposition in East Asia using the Community Multi-Scale Air Quality (CMAQ) model for 2001. By reducing each regions' $SO_2$, $NO_x$, and primary particle emissions, they demonstrated that foreign emissions contribute around half of total nitrogen deposition in South Korea. Using CMAQ
simulation of January 2007, Koo et al. (2008) demonstrated that $PM_{10}$ transport from China to Korea is significant, contributing up to 80 % of total concentration during the studied period. Using the CMAQ Decoupled Direct Method (DDM), Kim et al. (2016) also concluded that $PM_{10}$ concentration in Seoul is mostly contributed from Chinese industrial and urban regions (39.77 %–53.19 %), emissions in South Korea (15.37 %–37.10 %), and emissions in North Korea (9.03 %–18.05 %). All these studies, however, have examined short, episodic periods and/or have used a limited set of modeling configurations.



Therefore, the robustness of the estimated contributions have rarely been tested through a case estimating the partial contribution of foreign emission sources.

While many studies have addressed sources of uncertainties in the estimation of contributions or source apportionment, few have tried to investigate, whether qualitatively or quantitatively, the uncertainties resulting from the meteorological model.

Even though the chemical formation and transport of PM is very sensitive to the meteorological field (Akyüz and Cabuk, 2009), its impacts on the estimates of regional contributions are rarely studied. In this study, we estimate the contributions made by regional emission sources, domestic and foreign, to the PM concentrations in the Seoul Metropolitan Area (SMA), South Korea, discussing a range of uncertainties resulting from meteorology and emission inventories in order to assess the robustness of the estimated contributions. Section 2 introduces two modeling systems, the emissions inventories, and

observational data. Section 3 describes the contribution estimation method using a brute-force approach. Seasonal and spatial variations of meteorology and simulated surface PM, along with contributions from regional emissions, are shown in Sect. 4. Finally, Sect. 5 summarizes and discusses the findings.

## 2 Models & Observations

### 2.1 Modeling domain

In order to estimate the contributions made by South Korean emission sources (hereafter domestic emissions) and by other Asian countries besides South Korea (hereafter foreign emissions) to the surface concentration over the SMA, South Korea, we used two regional air quality modeling systems for East Asia. Simulations are designed to cover East Asia with 27-km grid resolution. Figure 1 shows the spatial coverage of the study; a 27-km domain over East Asia is shown in the left panel. An enlarged map of South Korea and the SMA (i.e., Seoul, Incheon, and Gyeonggi-do) are shown in the right panel.

**2.2 WRF-CMAQ system**

The Integrated Multi-Scale Air Quality system for Korea (IMAQS-K) is an air-quality forecast system over East Asia and Korea developed by Ajou University, Korea. Operational since May 2012, it now provides nine combinations of model configurations. For this study, we adopted a Weather Research and Forecasting Model (WRF) (Skamarock and Klemp, 2008)–Sparse Matrix Operator Kernel Emission (SMOKE)–Community Multiscale Air Quality (CMAQ) (Byun and Schere,

2006) modeling system to simulate gas and aerosol concentrations in East Asia. WRF version 3.3 was used for meteorology simulation, initiated with the National Center for Environmental Protection (NCEP) Global Forecast System (GFS) 0.5° × 0.5° global product. The Shuttle Radar Topography Mission (SRTM) Digital Elevation Model (DEM) with 90-m resolution and the Korean Ministry of Environment Land Use/Land Cover data are used for terrain data and surface land type data, respectively. The Meteorology–Chemistry Interface Processor (MCIP) (Otte and Pleim, 2010) version 3.6, is used as a

preprocessor for CMAQ simulation. CMAQ version 4.7.1, with the AERO5 aerosol module and Statewide Air Pollution





Research Center version 99 (SAPRC99; Carter, 1999), is used as the chemical mechanism in the chemical-transport modeling system.

## 2.3 UM-CMAQ system

The second modeling combination is a Unified Model (UM)-SMOKE-CMAQ framework. The UM is a weather and climate
modeling system from the United Kingdom Met Office, capable of simulating a wide range of spatial and temporal scales.(Price et al., 2011) The UM system, adopted by the Korea Meteorological Administration (KMA), has been an operational Numerical Weather Prediction system with 25 km global forecast resolution. Also, combined with the Asian Dust Aerosol Model (ADAM), it has been an operational system forecasting haze episodes (Lee et al., 2012). The UM is designed to solve non-hydrostatic, deep-atmosphere dynamics using a semi-implicit, semi-Lagrangian numerical scheme
(Cullen et al., 1997; Davies et al., 2005). The model uses an equatorial latitude–longitude horizontal grid, which rotates the North Pole to [306.97˚, 52.43˚] with Arakawa C staggering. Vertical coordinates are based on terrain, following hybrid-height with Charney–Phillips staggering (Davies et al., 2005). Physical options for surface scheme, boundary layer, cloud microphysics, and convection for the UM simulation are summarized in Table 2. Meteorological data are processed using UM-MCIP version 2.0, which was developed by Aeolus, the National Centre for Atmospheric Science (NCAS), and the UK
Met Office (http://cms.ncas.ac.uk/wiki/ToolsAndUtilities/UMMCIP). The UM-MICP is the interface between the Meteorology Chemistry Interface Processor and UM data, allowing the CMAQ atmospheric transport model to use directly data produced by the UM model.

## 2.4 Emissions inventories

Combinations of emissions inventories are used in the chemistry-transport model simulation to investigate the model's
sensitivity to meteorology and emission inventories. Table 1 summarizes the choices of emissions inventories for each model configuration. To isolate the impact of meteorology (M1 and M2), an identical emissions inventory set is used for all simulations. For anthropogenic emissions, the Intercontinental Chemical Transport Experiment–Phase B (INTEX-B) 2006 emissions inventory and the Model Inter-Comparison Study for Asia (MICS-Asia) emissions inventory are used for all Asian countries besides South Korea. Inside South Korea, emissions are replaced with those from the Clean Air Policy Support
System (CAPSS) 2007 and 2010 emissions inventory. In addition, four combinations of domestic and international emissions inventories are used to investigate the sensitivity of the model to the selection of emissions inventory.

**INTEX-B 2006**. The INTEX-B emissions inventory was developed by the National Aeronautics and Space Administration (NASA) to support the INTEX-B field campaign. Emissions from all major anthropogenic sources are included, while natural emissions sources, biomass burning, and dust are excluded. The 2006 INTEX-B emissions inventory provides
emissions for eight species—(1) $SO_2$; (2) $NO_x$; (3) CO; (4) non-methane volatile organic compounds (NMVOC); (5) $PM_{10}$; (6) $PM_{2.5}$; (7) BC; and (8) OC—with 30 min x 30 min spatial resolution over 22 counties and regions in Asia. Four emissions sectors—(1) power plants; (2) industry; (3) residential; and (4) transportation—are included in the data set. Data are



available at http://mic.greenresource.cn/intex-b2006 (Zhang et al., 2009). For the year 2006, total Asian anthropogenic emissions are estimated at 47.1 Tg $SO_2$, 36.7 Tg $NO_x$, 298.2 Tg CO, 54.6 Tg NMVOC, 29.2 Tg $PM_{10}$, 22.2 Tg $PM_{2.5}$, 2.97 Tg BC, and 6.57 Tg OC.

**MICS-Asia 2010.** The MICS-Asia emissions inventory was designed to support the MICS-Asia model inter-comparison studies and the Task Force on Hemispheric Transport of Air Pollution (TF HTAP) project. Emissions inventories were built to estimate all major anthropogenic sources in 30 countries and regions in Asia. Data from different regional emissions inventories are compared, with the best available from each region incorporated into a mosaic inventory at uniform spatial and temporal resolution. Total Asian emissions of ten species in 2010 are estimated at 51.3 Tg $SO_2$, 52.1 Tg $NO_x$, 336.6 Tg CO, 67.0 Tg NMVOC, 28.8 Tg $NH_3$, 31.7 Tg $PM_{10}$, 22.7 Tg $PM_{2.5}$, 3.5 Tg BC, 8.3 Tg OC, and 17.3 Pg $CO_2$. Monthly gridded emissions are provided at a spatial resolution of $0.25° \times 0.25°$ at http://www.meicmodel.org/dataset-mix (Li et al., 2015).

**CAPSS 2007 & 2010.** The CAPSS provides South Korean emissions and includes point, area, on-road, and non-road emissions sectors. It classifies sources of emissions into four levels, with 12 upper-level categories, 54 intermediate-level categories, 312 lower-level categories, and 527 detail-level categories. The upper-level categories include combustion in energy industries (point), non-industrial combustion plants (point & area), combustion in manufacturing industries (point & area), production processes (point & area), storage and distribution of fuels (point & area), solvent use (mobile), other mobile sources and machinery (mobile), waste treatment and disposal (point), agriculture (area), other sources & sinks (area), and fugitive dust (mobile & area). CO, NOx, SOx, $PM_{10}$, and VOCs emissions are provided for each upper-level category except for fugitive dust emissions (Lee et al., 2011).

**Natural Emissions.** The Model of Emissions of Gases and Aerosols from Nature (MEGAN; Guenther et al., 2006) is used for biogenic emissions, consideration of which is noteworthy. Estimation of biogenic emissions depends on its input to meteorological conditions (Guenther et al., 2006). Therefore, estimated biogenic emissions may differ with meteorological input. In this study, for both simulations we decided to use an identical biogenic emissions set, estimated using WRF model outputs with MEGAN, since we are focusing on the impact of differences in pure meteorological model in estimating foreign emissions. This decision might impact secondary PMs, but it has little impact on the primary sources of PM emission. Natural emissions from dust and wildfires are not included.

## 2.5 Observational data

Surface $PM_{10}$ concentrations and meteorological observations (e.g., 2-m temperature, 10-m wind speed, surface pressure, precipitation, and cloud fraction) are obtained respectively from the National Institute of Environmental Research (NIER) and Korea Meteorological Administration (KMA). Figure 1 shows the locations of surface monitoring sites. Locations of 102 SMA NIER surface-monitoring sites are also marked with black dots.



## 3 Methodology

Quantitatively estimating the contributions of local and neighboring emissions is crucial to the public, especially to inform policy meant to regulate emissions. Though the question of responsibility for regional air pollution is important, what is more important for policy-making is to identify the most efficient method of emissions regulation, which can sometimes be a very sensitive political issue. Emissions regulations hugely impact local economies, especially in rapid-developing countries. Better understanding of how each emissions source contributes to overall air quality helps to maximize the efficiency of emissions regulation, both domestically and internationally, with as little damage to the current economy as possible accompanied by maximum environmental improvement.

Studies of the long-range transport of pollutants and their impact on receptor regions have been conducted with numerous Lagrangian and Eulerian models. In the context of the source-receptor relationship, Positive Matrix Factorization (PMF) using the Potential Source Contribution Function (PSCF) or Chemical Mass Balance (CMB) have been popular approaches associated with Lagrangian trajectory models. Three-dimensional chemical transport models (CTMs) have also been used in simple brute-force method (BFM) approaches (Burr and Zhang, 2011; Itahashi et al., 2015), as well as in more complicated source-apportionment analyses based on DDM or High-order DDM (Cohan et al., 2005; Dunker et al., 2002; Koo et al., 2009; NAPELENOK et al., 2006).

To estimate the sizes of the impacts of domestic and foreign emissions to surface PM concentrations over South Korea, we have conducted two CMAQ simulations with brute-force emissions adjustments. In BFM, a common approach to analyze the response of modeled PM to changes in input emission, a model is run repeatedly with perturbed emissions, and the two simulation results are compared. This method is simple and popular for analyzing the contributions of local and/or regional emissions, but it has some limitations. First, theoretically, the sum of all source contributions does not necessarily equal the simulated concentrations in the base case if the model's response to emission inputs is not linear (Koo et al., 2009). Second, like most approaches involving simulation, the method is sensitive to the model's performance, especially regarding the transport of pollutants and their precursors, which is one of the main focuses of this study. Apparently, the BFM has its own limitations, mostly due to the nonlinear response characteristics of its final simulation to changes in emissions compared to mass conservative chemical-source apportionment methods, such as particulate matter source apportionment technology (PSAT) (Yarwood et al., 2007). However, this method still provides a compromising approach. Some methods are too simple and do not include chemical reactions, such as the formation of secondary aerosols, while other methods are too complicated, taking many more resources for larger and longer-term simulations.

We have conducted four cases with different meteorology (i.e., WRF and UM) and emission scenarios (base case and 50 % reduction):

    (1)   WRF-CMAQ base case using INTEX-B and CAPSS emissions;

    (2)   WRF-CMAQ BFM using INTEX-B and 50 % reduction of CAPSS emissions;

    (3)   UM-CMAQ base case using INTEX-B and CAPSS emissions; and



(4)  UM-CMAQ BFM using INTEX-B and 50 % reduction of CAPSS emissions.

Contributions of domestic and foreign emissions are estimated as:

$$Domestic\ Contribution = \frac{(C_{base} - C_{\Delta E})/\Delta E}{C_{base}} \times 100\ \% \ , \qquad (1)$$

$$Foreign\ Contribution = 100 - (Domestic\ Contribution)\ , \qquad (2)$$

where C is surface $PM_{10}$ concentrations and $\Delta E$ is the ratio of the emissions reduction test. Here, we used a 50 % reduction in South Korean emissions as a test.

Instead of calculating the theoretical zero-out-contribution of foreign emissions by removing total emissions without South Korean emissions, we conducted a sensitivity run in which we reduced South Korean emissions by 50 %. By doing so, we intended to minimize the chance of generating an unrealistic chemical environment. Since foreign emissions, especially those from China, clearly dominate domestic emissions, we decided to adjust the smaller portion (i.e., South Korean emissions) to provide a consistent chemical environment throughout all modeled scenarios. After several sensitivity tests, we selected a 50 % South Korean emission reduction scenario to compromise between the deficiencies of nonlinear chemical response and the risk of an extreme, poorly modeled chemical environment. The nonlinearity of modeled surface PM concentrations to the emissions reduction method is low, especially for $PM_{10}$. We assume that its response is mostly linear because its larger portion comes from primary emission sources.

## 4 Results

### 4.1 Model evaluation

Figure 2 summarizes the model performance for a meteorological simulation using 2-m temperature, 10-m wind speed, surface pressure, precipitation, and cloud fraction over the SMA, Korea. Dark gray lines (and black circles) indicate observations from KMA surface-monitoring sites, while blue and red lines represent simulated results from WRF and UM, respectively. Simulated daily average surface temperatures agree well with observations for both meteorological models, WRF and UM. The intensity of the wind field is one important difference. For most days in 2014, both WRF and UM show much stronger wind speeds than the measurements. Surface pressure agrees well with observation, except for an apparent offset: due to relatively coarse domain-grid resolution, the model's terrain cannot resolve the altitude of individual monitoring sites. For air-quality simulation, the timing of precipitation is important, but accurate evaluation of precipitation amounts is beyond the scope of this study. It should be carefully noted that this study does not intend to compare meteorological performance between models. Meteorological models should be evaluated with equitable evaluation metrics. Comparisons in this study instead frame a range of possible uncertainty for meteorological variables, which might affect chemical simulations undertaken as part of more ensemble approaches.



A comparison of cloud fractions between observations and the models also shows considerable discrepancy. Evaluation of cloud fields is challenging, based in part on the need to define so-called "cloudiness." Kim et al. (2015) found no consensus regarding the physical interpretation of clouds from ground observation, chemistry models, and satellites measurements. However, cloud fraction might have an important impact on photochemical reactions, such as the production of surface ozone. The impact of cloudiness on the formation of PM, however, is not fully understood, except for sulfate formation under cloudy conditions, especially in a highly polluted area like East Asia. Clearly, further studies are necessary to determine the direct and indirect impacts of cloud fraction. Thinking of it as an indicator of relative humidity and the possibility of precipitation may explain the relationship between surface PM and cloud fraction, because high relative humidity implies a higher chance of precipitation, which is a very effective means of wet-scavenging aerosols. Therefore, lower cloud fractions may be associated with higher PM concentrations. For both wind speed and cloud fraction, the UM-CMAQ system tends to produce higher PM concentrations.

On the other hand, the impact of wind velocity on surface PM concentration is straightforward, as low wind speeds are critical in generating stagnant conditions. High wind speed easily wipes out pollutants and their precursors. Wind may change not only total PM concentration, but also PM component fractionation. Kim et al. (2005) showed that changes in wind speed can significantly influence relative PM distribution patterns among coarse and giant particle fractions. Slight underestimations of surface temperature during springtime in both simulations are also interesting and may be important to the photochemistry of the surface zone, but we have no clear correlations in this study between these underestimations and modeled surface PM concentrations.

Figure 3 shows the spatial distribution of modeled surface $PM_{10}$ for each season. Both GFS-WRF-CMAQ and UM-CMAQ simulations show very similar spatial patterns, which makes sense, since both simulations use an identical emissions inventory. Both models show higher PM concentrations over China, especially during wintertime, as would be intuitively expected from the use of fossil fuels for residential heating and power plants. PM distribution over Korea is relatively high, but it is not easy to separate signals from China from those of Korea due to the intensity of Chinese pollutants and their transport during wintertime within the base simulation. Compared to China and Korea, Japan shows much reduced distribution of PM concentration. In the summer, PM concentrations over China are much weaker compared to other seasons, so it is possible to recognize local signals from Korea. Figure 3 also presents several interesting features that might require further investigation. The UM-CMAQ simulations have produced enhanced surface PM concentrations over the northern Pacific. Since the UM-CMAQ system shows a much weaker wind field compared to the WRF-CMAQ system, it is unlikely this is due to the transport of continental pollutants. Further investigation of the physical and chemical mechanism of the enhanced surface ozone over the northern Pacific would be an interesting future topic of study.

Figure 4 shows time series of daily averaged surface $PM_{10}$ concentrations observed at 102 surface monitoring sites in the SMA. Spatially and temporally collocated modelled concentrations from the WRF-CMAQ and UM-CMAQ systems are also shown in blue and red, respectively. Both simulations well reproduce general variations of SMA surface PM concentrations, but they also display clear limitations in several episodes. One typical problem in the chemical modeling of surface PM in





the SMA is that simulated surface PM concentration constantly underestimates observed measurements. At this point, we have no clear evidence for the reason models, including those in the current study, consistently underestimate SMA surface PM. Both models here failed to reproduce peaks, such as those on 21–22 January, 16 April, 23–24 April, 27–28 May, and 2 December, for example. In general, surface PM concentration simulated using UM-CMAQ generates higher PM

concentrations compared to the WRF-CMAQ system, which we suspect results from UM-CMAQ's weaker wind field, which results in a more stagnant and shallower boundary layer. However, the UM-CMAQ system also shows sporadic overestimations, such as 14–16 August, 31 August, and 12 September.

## 4.2 Contribution estimation

Contributions from domestic and foreign emissions sources to surface PM concentration over the SMA are estimated using

the BFM method described in Sect. 3. Figure 5 shows time series during 2014 of daily averaged partial contributions from domestic and foreign sources. Foreign contributions are shown in red, while domestic contributions are shown in blue. Though both modeling systems display similar temporal variation, their relative distributions have clear seasonal variation. Figure 6 summarizes the seasonal variation of estimated contributions from foreign emissions to the surface concentration of $PM_{10}$ in the SMA region using the WRF-CMAQ (blue) and UM-CMAQ (red) systems. Grid cells in the SMA region are

selected using global administrative boundary GIS data, and their averages, first quartiles, and third quartiles are shown in a whisker plot. In general, the estimated contributions of foreign emissions have similar seasonal variations, with higher contributions during cold seasons and lower contributions during the summer season. The peaks of the estimated foreign emissions contributions are slightly different, however; the WRF-CMAQ system estimates the highest contribution in March, while the UM-CMAQ system estimates the highest contribution in December.

Interestingly, the estimated contribution is relatively low in May, which is typically known as a season with more long-range transport, as would be predicted by the increased wind speed. This could imply that the intensity of wintertime emissions from source regions (e.g., residential heating from northern China) plays a more important role than this enhanced transport during springtime.

In summer, beginning in June, the estimated foreign emissions contributions drop quickly, for which we may consider two

factors as explanation. First, in June, when the Asian Monsoon starts, wet scavenging of pollutants becomes more efficient, so conditions are not favorable for the long-range transport of pollutants and their precursors. Second, during the summer season, with the northward move of the Intertropical Convergence Zone, the wind direction reaching the SMA region is typically southerly or southwesterly compared to the westerly or northwesterly flow in spring. The typical transport pathway changes, then, from one through northern China, which is the strongest source of pollutants, to one through southern

China.(Kim et al., 2014) Further studies using detailed regional source locations are required to address the contributed portions from each region in more depth.

Figure 7 shows the spatial distributions of foreign emissions contributions to each major city and province in South Korea. Consistent with the time series analysis in Figure 6, these have clear seasonal variation, with higher contributions during the





cool season and lower contributions during the summer season. During December–January–February (DJF) and March–April–May (MAM), foreign emissions are dominant over most of the Yellow Sea, while domestic emissions affect regions over the East Sea. The impact of foreign emissions offshore of Ulsan and Busan cities are especially weaker (i.e., meaning stronger domestic impact), reflecting the impact of this highly industrialized area in southeastern Korea. Also noticeable is

that major cities (e.g., Seoul, Busan, Daegu, Incheon, Daejeon, Gwangju, and Ulsan) are affected less by foreign emissions; this makes sense because local emissions are more dominant in urban or industrialized regions, with transported pollutants or precursors contributing less to the local surface PM concentrations. Monthly contributions from foreign emissions sources as estimated by each model are listed in Table 5 and Table 6 for each administrative boundary region.

**4.3 Sensitivity to meteorology**

Both meteorology models, GFS-WRF and UM, simulate meteorological conditions reasonably well in terms of daily and annual variations and the development and passage of synoptic systems. The most noticeable differences between two systems are wind speed and cloud fraction. The impact of wind speed on surface PM concentration is straightforward; usually, more stagnant conditions are critical to produce high pollutant concentrations. The impact of cloud fraction, however, is unclear and not yet fully studied. Likely possibilities include the formation of sulfate under cloud conditions, but

a more quantitative explanation of the impact of cloudiness on surface PM concentration remains outstanding.

This study clearly shows that accurate wind field simulation in regional air quality simulations is important, from two perspectives. First, the intensity of wind is critical to generating calm or stagnant conditions. High-pollutant events require stagnant wind conditions, because high wind speeds easily wipe out pollutants. In the current study, the UM-CMAQ system predicts higher PM concentrations even though an identical emissions inventory is used in both simulations. Second, the

timing of frontal passages is critical in the dissipation stage of high-pollutant events, as highlighted in the case study of the February 2014 episode, discussed below. Wind fields might control the absolute concentrations, component fractions, and remote source contributions of local surface PM levels.

One recent type of episode over East Asia of serious public concern is the occurrence of extraordinarily long-lasting episodes of high PM concentration. Figure 8 shows a particular case, a high PM episode in February 2014 during which a stagnant

high-pressure system stayed over the Yellow Sea for more than a week, causing severe haze events in both China and Korea (Kim et al., 2016b). Figure 8 shows time series of $PM_{10}$ in the SMA during this high PM episode, from 22 February to 2 March. Both models simulate PM concentrations reasonably well. On 20 February, PM levels reached as low as 50 μg m$^{-3}$. Both models well simulate the gradual build-up of surface PM level through 24 February, although the UM-CMAQ system overestimates from 22 February to 24 February. After holding around 150 μg m$^{-3}$ for days, the biggest difference, however,

comes at the end of the high PM event. The WRF-CMAQ system simulates the intensity of frontal passage activity on 28 February too strongly, resulting in a quick sweep out of surface PM concentration, while observations still show enhanced PM concentration level around 80–100 μg m$^{-3}$. The UM-CMAQ system better simulates these changes, showing a slow dissipation of the high PM episode.



From 20 February to 2 March, the mean of observed PM concentration was 106.5 µg m$^{-3}$, with the highest concentration of 195.9 µg m$^{-3}$ on 26 February. Modeled mean concentrations are 75.5 µg m$^{-3}$ and 112.4 µg m$^{-3}$ and estimated foreign emissions contributions are 81.3 % and 81.7 % from the WRF-CMAQ and UM-CMAQ systems, respectively. This is an interesting case showing that, while simulated absolute surface PM concentration is very sensitive to meteorology, the modeled relative contribution of different sources is still very robust. This severe haze case was one of the worst in recent years; more detailed analyses of its chemical composition and regional contributions are discussed elsewhere (Kim et al., 2016a).

## 4.4 Sensitivity to emissions inventories

The models' sensitivities to emissions inventory are more complicated than their sensitivity to meteorological field, because not only the total amount of emissions but also the total and relative amount of each primary PM and precursor emissions must be considered. In this section, we consider the range of uncertainty in the modeled estimation of regional contributions resulting from the choice and configuration of international and domestic emissions inventory sets. Detailed discussion of the impact of each PM species will be provided in a forthcoming study (Bae et al., 2016).

Figure 9 shows the seasonal variation of foreign emissions contributions to surface PM$_{10}$ concentrations over the SMA and South Korea. As described in Table 1, these four simulations (E1–E4) use different selections of emissions inventory. All cases generally agree in terms of seasonal variation, showing higher contributions of foreign emissions during the cold season and lower contributions during the summer season.

The biggest difference is between E1 and the other cases. While estimated foreign contributions in the E1 simulation also show similar seasonal variation, their estimated contribution is much lower than in the other simulations, with the E1 case implying that local emissions sources are more dominant over surface PM concentration. This difference results from differences in the amount of primary PM emissions in the SMA, as summarized in Table 3. Primary emissions assigned to the SMA from the INTEX-B 2006 emissions inventory, 90,858 TPY, are two to three times higher than emissions from the CAPSS 2007 or 2010 inventories. At this point, we cannot conclude which combination of emission inventories (i.e., E1–E4) provides the most realistic simulation of PM in the SMA, mostly because there are no nationwide measurements of PM speciation.

Figure 10 plots simulated surface PM$_{10}$ concentrations and estimated foreign contributions over the SMA. In all cases, high surface PM concentration in the SMA is well associated with high contributions from foreign emissions sources. For the E2–E4 simulations, foreign emissions contributions reach as high as 70–80 % when monthly mean surface concentrations are over 30 µg m$^{-3}$, while estimated contributions of foreign emissions are as low as 40 % when SMA PM$_{10}$ concentrations are lower than 25 µg m$^{-3}$. The E1 case has a slightly lower regression slope between PM concentration and foreign contribution, but it still shows a prominent correlation (i.e., R=0.75).





## 5 Conclusion

This study assesses contributions from domestic and foreign emissions sources to surface PM concentrations in the SMA, Korea. Located downwind of dominant sources of anthropogenic emissions, South Korea has experienced direct and indirect impacts of transported pollutants and their precursors from foreign emissions sources. Quantitative estimation of the impact of each emission source are therefore crucial to the planning of emissions regulation policy.

Two regional air quality modeling systems, WRF-CMAQ and UM-CMAQ, were used to estimate the impact of local and remote emissions during 2014 using a brute force emission-adjustment approach. In the SMA region, the annual mean contributions from foreign emissions sources are estimated at 67.6 % and 63.6 % by the WRF-CMAQ and UM-CMAQ systems, respectively. Estimated foreign contributions show clear seasonal variation, comprising up to 80 % during the cold season and as low as 40 % during summertime. Changes in wind direction may play the biggest role in this seasonal change, with enhanced wet scavenging during summertime likely being unfavorable for the long-range transport of foreign emissions and pollutants. Estimated contributions also depended on geography: western coastal regions on the Yellow Sea are affected more by foreign emissions, implying a strong impact of Chinese emission sources. Major cities are usually less affected by foreign emissions as a percent of total share, since their local emissions are already stronger.

Simulations of surface PM concentration and estimated source contributions were found to be sensitive to the choice of meteorology and emissions inventories. Considerable differences in simulated surface PM concentrations likely resulted from differences in wind speed simulation between the WRF and UM systems. Estimated source contributions, however, differed less significantly by different meteorology, resulting in a more robust source contribution estimation. Since partial contributions are mostly determined by the relative strengths of pollutants from each region, estimated contributions from local compared to remote sources were not greatly affected. Simulations using multiple combinations of emissions inventories all showed similar seasonal variation. The largest difference between these combinations was found when the domestic emissions inventory had strong primary PM emissions (i.e., the E1 case). This result also confirms the importance of determining the balance of domestic and foreign emission sources when estimating relative contributions.

This study is intended to provide reference information regarding model uncertainty in meteorology simulation. We found that differences in meteorological model can lead to considerable, but not significant, differences in the estimation of contributions from regional (e.g., domestic and international) emissions sources. We conclude that the estimation method itself is valid, but the modeled results should be considered with caution when interpreted for emission regulation policy-making. Based on the findings from this study, our future study will focus on two topics: (1) the response and sensitivity of surface PM concentration and source contribution to changes in PM components; and (2) further evaluation of wind fields, especially across the Yellow Sea.





**Acknowledgements**

This research was partly supported by PM2.5 research center supported by Ministry of Science, ICT, and Future Planning (MSIP) and National Research Foundation (NRF) of Korea (NRF-2014M3C8A5030624), and National Institute of Meteorological Sciences for Research and Development for KMA Weather, Climate, and Earth system Services (NIMS-2016-3100).

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





**Table 1. Modeling configurations for meteorology and emissions inventories.**

| Cases | Meteorology | Emissions Inventory | |
|---|---|---|---|
| | | **Foreign** | **Domestic** |
| **M1** | WRF | INTEX-B 2006 | CAPSS 2007 |
| **M2** | UM | | |
| **E1** | | INTEX-B 2006 | INTEX-B 2006 |
| **E2** | WRF | INTEX-B 2006 | CAPSS 2007 |
| **E3** | | MICS-Asia 2010 | CAPSS 2007 |
| **E4** | | MICS-Asia 2010 | CAPSS 2010 |



**Table 2. Physical options for meteorological and chemical simulations.**

| Options | | Model | |
|---|---|---|---|
| | | WRF | UM |
| | Initial field | GFS | Global UM |
| | Microphysics | WSM3 (Hong et al., 2004) | Mixed-phase precipitation (Wilson and Ballard, 1999) |
| Meteorology | Cumulus scheme | Kain-Fritsch (Kain, 2004) | Modified Mass Flux Convection with CAPE Closure (Gregory and Rowntree, 1990) |
| | LSM scheme | NOAH (Chen and Dudhia, 2001) | MODES-II (Essery et al., 2001) |
| | PBL scheme | YSU (Hong et al., 2006) | First order non-local BL scheme (Lock et al., 2000) |
| | | CMAQ | |
| | Chemical mechanism | SAPRC99 (Carter, 2003) | |
| | Chemical solver | EBI (Hertel et al., 1993) | |
| | Aerosol module | AERO5 (Binkowski, 2003) | |
| Chemistry | Advection scheme | YAMO (Yamartino, 1993) | |
| | Horizontal diffusion | Multiscale (Louis, 1979) | |
| | Vertical diffusion | Eddy (Louis, 1979) | |
| | Cloud scheme | RADM (Chang et al., 1987) | |



**Table 3. Summary of NOx, SO2, NH3, and PM emissions for INTEX-B 2006, MICS-Asia 2010, and CAPSS (2007 & 2010) emissions inventories.**

|  | China | South Korea | SMA |
|---|---|---|---|
| NO$_x$ emissions; Unit: Tons per Year (TPY) | | | |
| INTEX-B 2006 | 19,347,446 | 928,703 | 303,627 |
| MICS 2010 | 27,267,065 | | |
| CAPSS 2007 | | 1,154,401 | 324,834 |
| CAPSS 2010 | | 1,012,476 | 232,387 |
| SO$_2$ emissions (Unit: TPY) | | | |
| INTEX-B 2006 | 26,536,326 | 465,273 | 102,654 |
| MICS 2010 | 27,162,387 | | |
| CAPSS 2007 | | 386,133 | 35,206 |
| CAPSS 2010 | | 380,103 | 20,076 |
| NH$_3$ emissions (Unit: TPY) | | | |
| INTEX-B 2006 | 12,395,592 | 179,668 | 33,515 |
| MICS 2010 | 9,956,950 | | |
| CAPSS 2007 | | 309,947 | 56,510 |
| CAPSS 2010 | | 267,027 | 35,832 |
| Primary PM emissions (Unit: TPY) | | | |
| INTEX-B 2006 | 16,111,075 | 332,586 | 90,858 |
| MICS 2010 | 16,982,600 | | |
| CAPSS 2007 | | 182,728 | 32,815 |
| CAPSS 2010 | | 345,693 | 45,688 |



**Table 4. Observed and modeled surface PM10 concentrations over surface monitoring sites in the SMA, South Korea. Underlined values indicate maxima and minima; unit is μg m-3.**

|  | Jan | Feb | Mar | Apr | May | Jun | Jul | Aug | Sep | Oct | Nov | Dec | Annual |
|---|---|---|---|---|---|---|---|---|---|---|---|---|---|
| Observed | 60.59 | 62.56 | 63.96 | 63.61 | 67.93 | 44.13 | 43.29 | 34.56 | 34.3 | 38.52 | 49.5 | 48.35 | 50.94 |
| WRF-CMAQ | 38.2 | 40.7 | 41.77 | 38.74 | 38.09 | 24.85 | 27.81 | 22.35 | 24.64 | 23.71 | 35.65 | 31.74 | 32.36 |
| UM-CMAQ | 43.8 | 62.36 | 51.09 | 43.67 | 36.43 | 37.02 | 36.98 | 37.46 | 41.59 | 28.21 | 40.84 | 29.59 | 40.75 |



**Table 5. Summary of foreign contributions to surface PM concentrations for each month over major cities and provinces in South Korea; unit is %.**

| Cities | Jan | Feb | Mar | Apr | May | Jun | Jul | Aug | Sep | Oct | Nov | Dec | Annual |
|---|---|---|---|---|---|---|---|---|---|---|---|---|---|
| Seoul | 56.7 | 72.8 | 73.1 | 61.4 | 62.4 | 42.2 | 37.3 | 35.4 | 39.6 | 36.2 | 59.1 | 55.6 | 55.3 |
| Busan | 61.1 | 67.6 | 68.6 | 58.7 | 55.4 | 30 | 33.7 | 42.9 | 35.2 | 48.7 | 47.2 | 64.2 | 52.8 |
| Daegu | 62.6 | 62.5 | 72.3 | 54 | 55.5 | 30.1 | 40.6 | 38.5 | 31.2 | 32.8 | 51.3 | 62.1 | 52.8 |
| Incheon | 78.6 | 82.7 | 88 | 73.1 | 80.9 | 55.1 | 52.3 | 48.5 | 53 | 54.1 | 74.3 | 76.8 | 71.5 |
| Gwangju | 68 | 63.7 | 76.5 | 55.4 | 69.8 | 37.4 | 50.9 | 31.9 | 22.9 | 32.5 | 54.4 | 67.9 | 56.5 |
| Daejeon | 69.5 | 67.9 | 73.8 | 53 | 67.8 | 32.3 | 48.3 | 30.1 | 32.5 | 39.3 | 60 | 64.5 | 57.6 |
| Ulsan | 56.6 | 66.6 | 67.1 | 51.8 | 48 | 22.3 | 24.6 | 35.8 | 29.8 | 40.7 | 44.5 | 62.1 | 47.1 |
| Gyeonggi-do | 71.3 | 79.2 | 81.4 | 67.4 | 65.8 | 45.7 | 42.7 | 42.9 | 48.8 | 46.8 | 70.2 | 69.2 | 64.4 |
| Gangwon-do | 74.6 | 85.1 | 83.5 | 71.1 | 74.7 | 49.4 | 51.4 | 60.1 | 66.5 | 50.6 | 71 | 80.9 | 71.4 |
| Chungcheongbuk-do | 70.6 | 70.9 | 76.6 | 59.4 | 65.7 | 34.4 | 46.8 | 39.8 | 42.1 | 40.8 | 62.6 | 73.2 | 60.9 |
| Chungcheongnam-do | 76.3 | 74.4 | 83.4 | 58.1 | 75.2 | 38.3 | 50.7 | 34.9 | 39.3 | 47.8 | 67.1 | 74.4 | 64.1 |
| Jeollabuk-do | 71.9 | 69.3 | 80.6 | 59.1 | 74.7 | 42.6 | 61.7 | 39.8 | 34.4 | 40.3 | 60.8 | 73.2 | 63.2 |
| Jeollanam-do | 76.4 | 67.4 | 79.8 | 58.4 | 73.3 | 43.9 | 55 | 43.8 | 33.4 | 44.7 | 58.8 | 76.6 | 62.5 |
| Gyeongsangbuk-do | 63.4 | 69.8 | 74.5 | 59.4 | 59.8 | 34.2 | 46.4 | 48.4 | 45.1 | 38.9 | 55.2 | 71.3 | 58.3 |
| Gyeongsangnam-do | 67.1 | 67.6 | 74.6 | 58.6 | 59.7 | 32.9 | 44.3 | 44.5 | 36.8 | 40 | 51.7 | 66.5 | 56.7 |
| Jeju-do | 109.2 | 77.1 | 90.3 | 87.8 | 100.8 | 79.5 | 105.2 | 88.8 | 75.4 | 84.6 | 82.1 | 99.4 | 90.3 |



**Table 6. Summary of foreign contributions to surface PM concentrations for each month over major cities and provinces in South Korea using the UM-CMAQ system; unit is %.**

| Cities | Jan | Feb | Mar | Apr | May | Jun | Jul | Aug | Sep | Oct | Nov | Dec | Annual |
|---|---|---|---|---|---|---|---|---|---|---|---|---|---|
| Seoul | 66.8 | 76.5 | 72.8 | 58.4 | 57.1 | 35.6 | 29.4 | 32.7 | 34.2 | 39.2 | 60.2 | 70.6 | 54.5 |
| Busan | 60.4 | 59.8 | 66.2 | 46.4 | 52.5 | 31.7 | 35.5 | 29.6 | 19.5 | 30.7 | 38.8 | 64.5 | 47.2 |
| Daegu | 61.7 | 65 | 64.2 | 43.8 | 47.9 | 20.7 | 26.6 | 17.8 | 17.4 | 26.4 | 41.9 | 64.7 | 46 |
| Incheon | 83.9 | 83.8 | 82.2 | 67.3 | 69 | 41.1 | 37.6 | 42.4 | 40.8 | 59.6 | 78.4 | 89.3 | 67.2 |
| Gwangju | 70.9 | 66.1 | 73 | 48.4 | 63.9 | 32.5 | 41.2 | 15 | 8.3 | 19.3 | 48 | 74.2 | 51.3 |
| Daejeon | 73.9 | 70.9 | 72.1 | 50.3 | 59.1 | 24.3 | 38.7 | 21.5 | 14.8 | 33.2 | 57.6 | 76.7 | 54.4 |
| Ulsan | 56.6 | 61.5 | 61.2 | 42.5 | 44.7 | 23.1 | 27 | 20.6 | 18 | 25 | 35.9 | 58.3 | 42.5 |
| Gyeonggi-do | 79.7 | 78.6 | 77.1 | 60.1 | 58 | 32.6 | 29.8 | 30.6 | 31.1 | 46.8 | 71.4 | 82.8 | 59.8 |
| Gangwon-do | 76.2 | 80.2 | 77.2 | 65.4 | 65.7 | 42.3 | 49.6 | 45.3 | 47.2 | 44.6 | 63.1 | 84.2 | 66.5 |
| Chungcheongbuk-do | 73.5 | 70.5 | 72.5 | 54.5 | 57.3 | 25 | 41.1 | 23.7 | 21.7 | 31.8 | 57.4 | 79.1 | 56.2 |
| Chungcheongnam-do | 80.1 | 75.9 | 79.5 | 56.2 | 67.5 | 29.6 | 41.3 | 25.2 | 21.5 | 45 | 64.8 | 84.7 | 59.9 |
| Jeollabuk-do | 75.1 | 70.5 | 76.2 | 54.4 | 68 | 34 | 49.8 | 24.3 | 15.5 | 27.1 | 53.9 | 81 | 58 |
| Jeollanam-do | 74.8 | 67.8 | 76.6 | 55.2 | 72 | 44 | 51.7 | 30.2 | 22.5 | 32 | 52.3 | 79.4 | 59.1 |
| Gyeongsangbuk-do | 65 | 71.6 | 70.2 | 55.7 | 59.9 | 35.8 | 44.9 | 34.6 | 35.1 | 31.5 | 46.4 | 72.7 | 56.7 |
| Gyeongsangnam-do | 67.2 | 65.2 | 70.3 | 51.6 | 60.5 | 32.6 | 38.8 | 27.5 | 18.8 | 24.5 | 42.3 | 71.6 | 52.1 |
| Jeju-do | 95.7 | 82 | 94.7 | 91 | 96.9 | 85.5 | 94.2 | 82 | 77.1 | 89.7 | 82.6 | 96.6 | 89.8 |





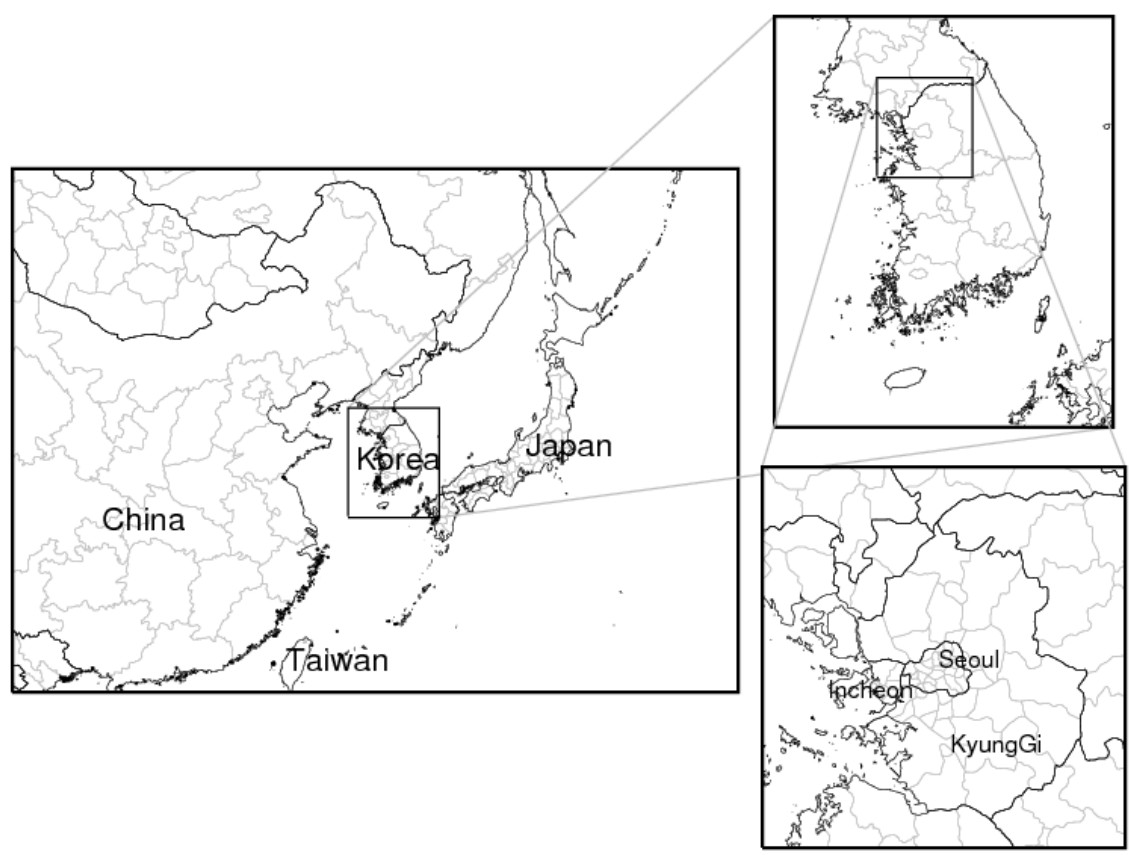

**Figure 1: Geographical extent of study domain: East Asia, South Korea, and the SMA.**





**Figure 2: Time series and scatter plots of meteorological components.**





**Figure 3: Spatial distributions of PM10 concentrations over East Asia using the WRF-CMAQ and UM-CMAQ systems, as marked.**



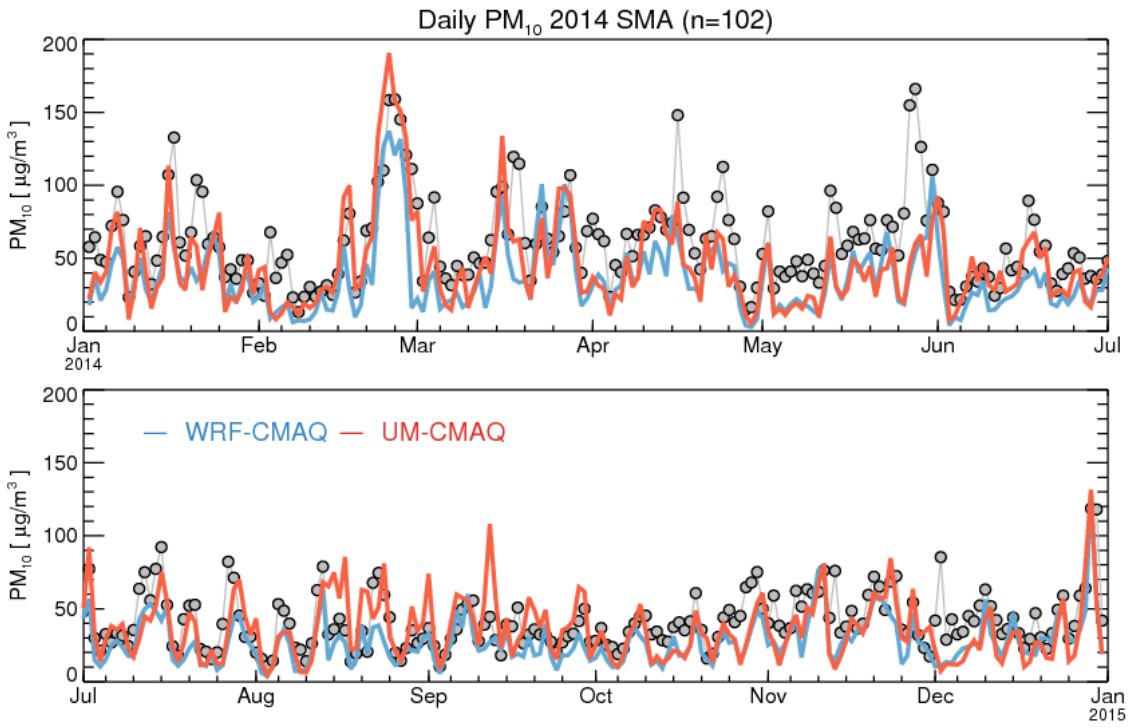

**Figure 4: Time series of daily averaged PM10 over SMA surface monitoring sites during 2014.**





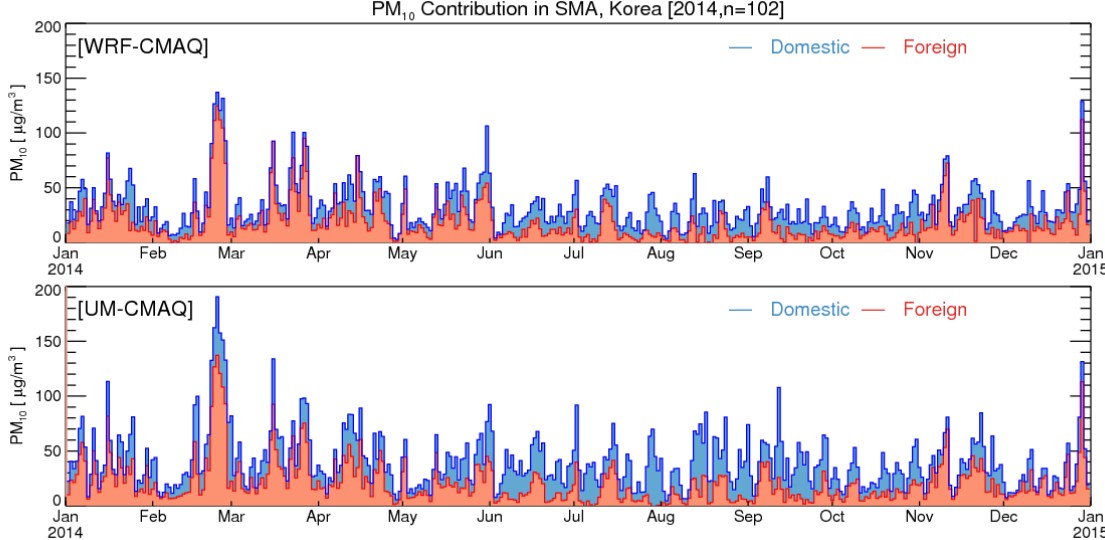

**Figure 5: Time series of contributions to surface PM10 concentrations in the SMA. Blue represents contributions from domestic emissions, while red represents contributions from foreign emissions. Contributions are calculated based on the 50 % brute-force reduction of South Korean emissions.**





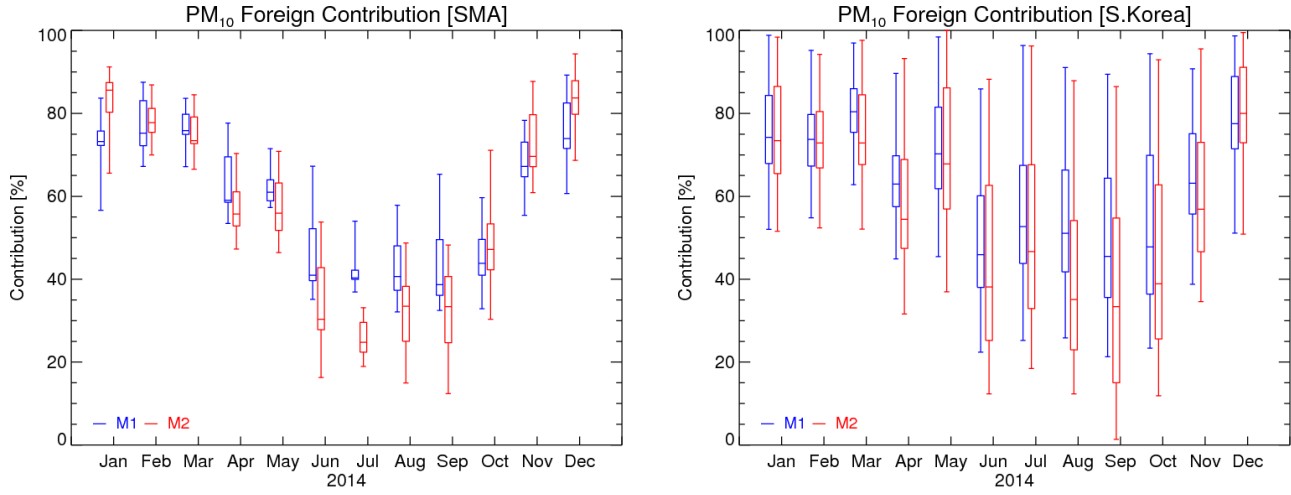

**Figure 6: Seasonal variation in foreign emissions contributions to the SMA (left) and to South Korea (right).**





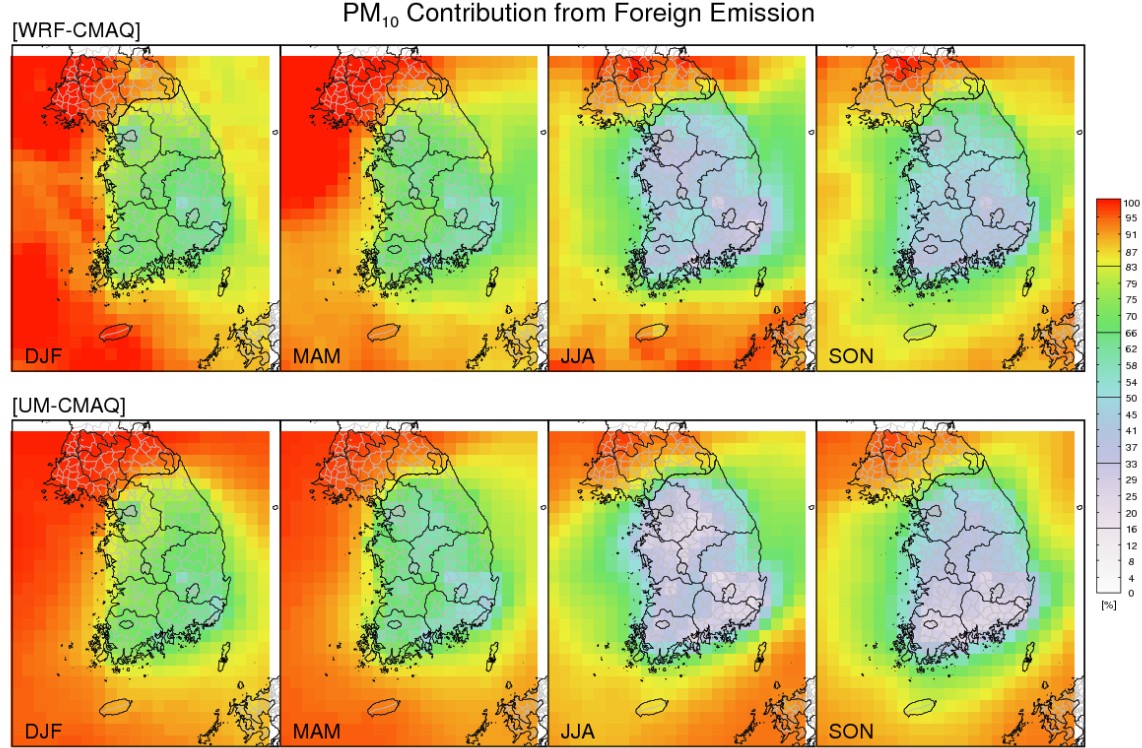

**Figure 7: Quarterly average foreign emissions contributions, as simulated by WRF-CMAQ and UM-CMAQ systems.**





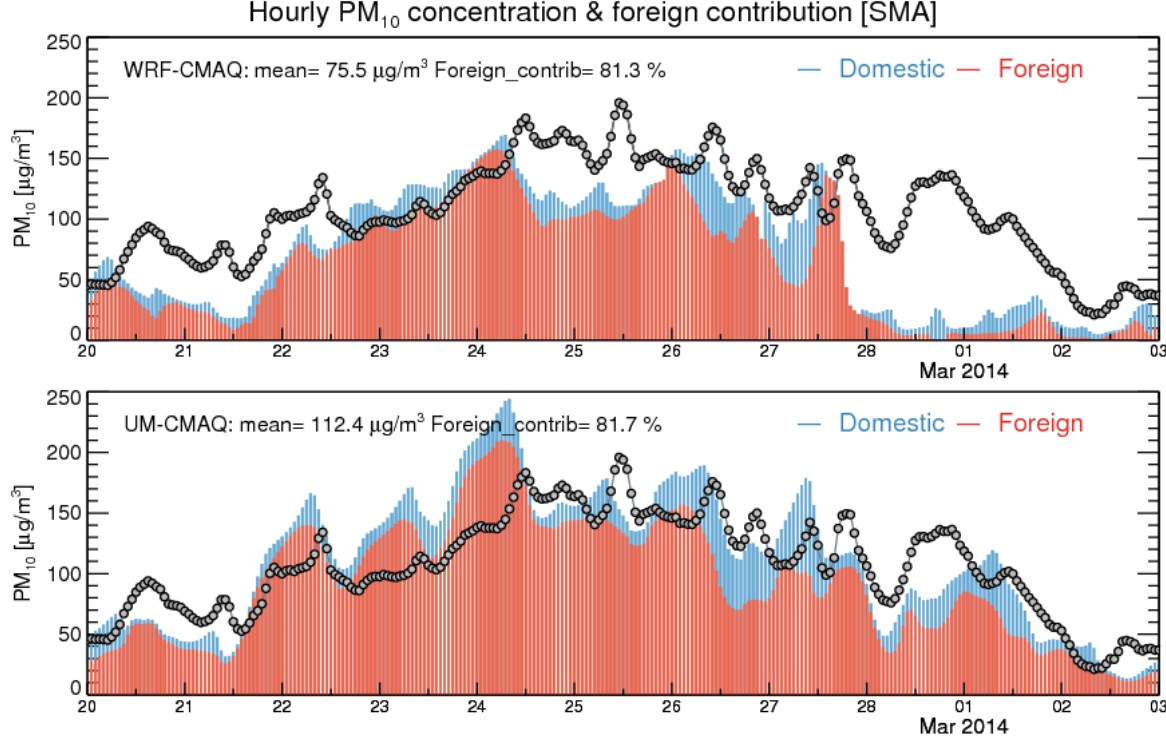

**Figure 8: Time series of surface PM10 concentration over the SMA from observations and models during the period from 20 February, 2014 to 3 March, 2014. Contributions from domestic and foreign emissions are shaded with blue and red, respectively.**





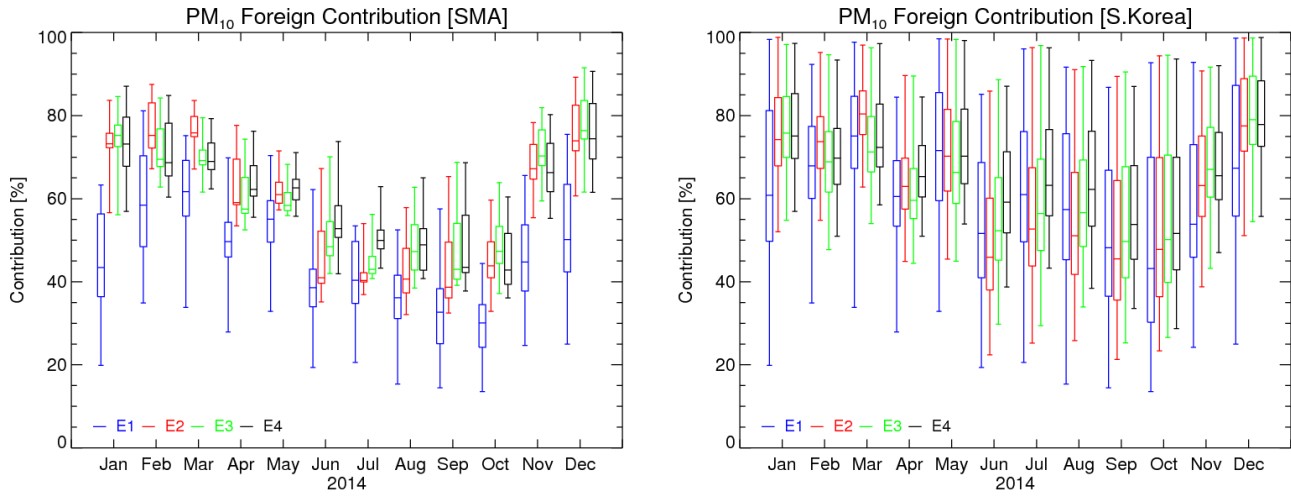

**Figure 9: Seasonal variations in foreign emissions contributions to surface PM10 concentrations over the SMA (left) and over South Korea (right). Simulation cases E1 – E4 indicate emissions inventory combinations as described in Table 1.**





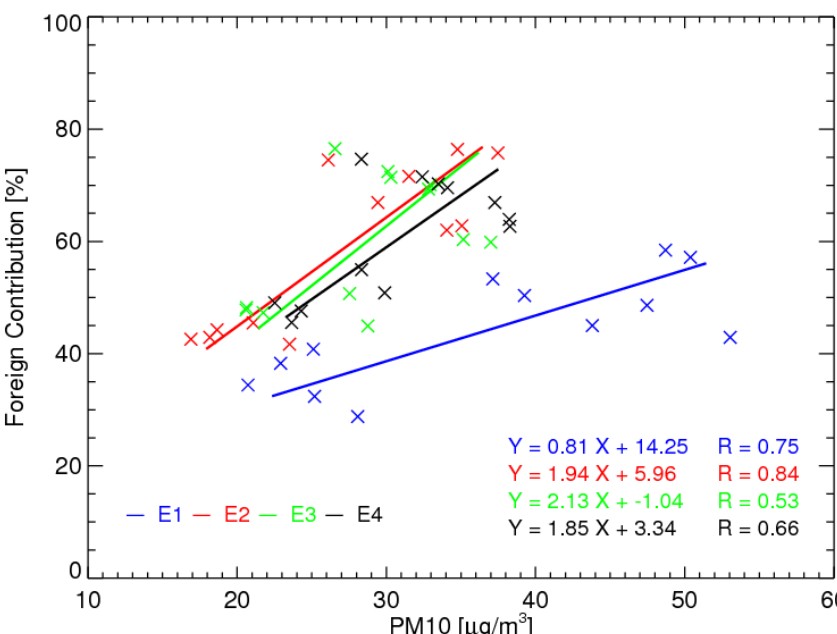

**Figure 10: Monthly mean SMA PM10 concentrations versus estimated contributions from foreign emissions sources.**