# Peer review of "Regional Contributions to Particulate Matter Concentration in the Seoul Metropolitan Area, Korea: Seasonal Variation and Sensitivity to Meteorology and Emissions Inventory"

_Atmospheric Chemistry and Physics, 2016_

## Referee Comment (RC1) · Anonymous Referee #1 · 28 Jan 2017

This manuscript seeks to distinguish between Korean and foreign contributions to particulate matter in Korea. It applies two meteorological models and four emissions cases to a full year episode to characterize how the source attributions differ across the cases. The modeling shows that the relative share of PM from domestic and foreign sources differs substantially by day and season.

Overall, most methods are sound, the article is well explained, and the figures and tables are clear. My main concerns are that: 1. The findings are framed as being robust and representing uncertainty. However, this limited number of cases does not

constitute a comprehensive ensemble or represent the range of uncertainty that may exist in the emissions inventory. The modeling relies on a somewhat arbitrary set of 4 emissions inventories, which differ for domestic and foreign sources and which do not directly represent the year (2014) that is simulated. Thus, it is not justified to call the findings robust (p. 1, line 25 and p. 12, line 19) and more caution is needed in interpreting the findings. 2. Source contributions are defined by doubling the impact of 50% emission reduction runs. However, if the reductions are applied only to anthropogenic emissions (this was unclear), then some of what is being termed "foreign" is actually resulting from biogenic emissions within Korea. Also, zero-out impacts are often larger than 2x the impact of 50% out cases, due to nonlinearities of the chemistry of pollutant formation. If that is the case here, it would systematically under-represent the domestic contribution, and hence over-represent the foreign share. One run should be conducted to test the linearity of response from 50% to 100% reduction, and language should be more cautious in defining source apportionment if it is based on 50% cases. 3. The model substantially under-estimates observed PM. This raises serious doubt about the conclusions, since it could indicate error in either the domestic or foreign emissions inventory.

Minor suggestions: p. 2, line 28: "region's" p. 2, line 33: excess precision in numbers p. 3, lines 3-6: Meteorological uncertainty has been studied elsewhere, with larger ensembles than considered here p. 6, line 26: "compromising" is the wrong word p. 9, lines 1-7: These explanations are not convincing, and the discussion of specific days is not helpful p. 12, line 25: How can results be "considerable but not significant"?

---

## Referee Comment (RC2) · Anonymous Referee #2 · 6 Feb 2017

This manuscript quantifies trans-boundary anthropogenic contributions to Korean modal particle distribution using brute-force sensitivity in annual simulations with two different mesoscale implementations of the CMAQ model driven by different meteorology and identical emissions inventories. The applied scientific value of the article is to set a realistic operational standard for the quality and quantity of evidence supporting spatial attribution of anthropogenic particle source contributions in East Asia, and the expected findings of substantial seasonal and sub-seasonal variability in foreign and domestic contributions that warrant annual simulations for such studies. While not a true ensemble, the two CMAQ implementations are both mature operational forecast-

ing systems, and they are truly different, with fundamental differences in simulated daily meteorology and chemical transport that add confidence to the similarity in source attribution. The article generally succeeds in what it attempts to achieve, limited mostly by the apparent absence of dust and wildfire emissions in all baseline and sensitivity simulations, the difference in vintage between emissions and simulation period, and the resultant low biases in simulated concentrations.

Major comment

The authors do not list a source for regional dust, wildfire, and biomass burning emissions, beyond the domestic fugitive dust from CAPSS, and only state that dust and biomass burning are excluded from the INTEX-B and MEGAN emissions inventories. The reader assumes this to mean no such emissions were employed, which is an obvious explanation for consistent low biases in simulated concentrations and pollution events, and a major limitation in the representation of primary and secondary PM and the linearity of emissions sensitivities. The authors must clarify the sources for these emissions in the regional and domestic inventories or clearly state that none were used. Their absence would obviously contradict the authors' claim that they "have no clear evidence for the reason models, including those in the current study, consistently underestimate SMA surface PM," especially after citing six sources on the impacts of dust storms on air pollution in the region. While their absence would not affect the attribution or linearity of anthropogenic sources to primary particle concentrations, it would substantively impact the resultant percentages, and would challenge the linearity in attribution for secondary particles and net particle concentrations. These are major limitations that may reduce the value of the results in applied decision support, even if the direct assessment of anthropogenic spatial attribution were otherwise accurate. Moreover, most of the value of the article for readers beyond Korea is in the transferable experimental framework for trans-boundary source apportionment, and the complete absence of known major emissions sources in the region limits the value of this study as representing a minimum operational standard for such assessments.

If it is the case that the simulations do not include dust and wildfire emissions, a minimum of one annual baseline simulation and one BFM sensitivity simulation in one of the models with such emissions would be warranted. Those additional simulations would then allow the authors to quantify the contributions of those sources relative to anthropogenic sources, quantify sensitivity and non-linearity under more realistic chemical conditions, quantify net international attribution, and instill confidence in simulations that otherwise exhibit profound and consistent low biases. If prior studies with one of the modeling systems have already sufficiently quantified these contributions for the same year, or one with similar dust and biomass burning emission, citation and summary might suffice. A second set of simulations to assess the role of transported biogenic emissions would be valuable, but less critical.

Technical comment

The authors do not define the simulation period until the results section. Specific start and end dates for the simulation and any initialization period should appear in the first paragraph of section 2.

---

## Author Comment (AC1) · 7 May 2017

**Authors' response to the review comments #2**

**"Regional Contributions to Particulate Matter Concentration in the Seoul Metropolitan Area: Seasonal Variation and Sensitivity to Meteorology and Emissions Inventory" by Kim et al.**

The authors express their appreciation to the two reviewers and the editor. We believe that their comments are very productive and substantially contributed to improving the manuscript. We offer general responses and point-by-point responses to the issues and comments addressed by the reviewers. Reviewers' comments are shown in italics.

*The article generally succeeds in what it attempts to achieve, limited mostly by the apparent absence of dust and wildfire emissions in all baseline and sensitivity simulations, the difference in vintage between emissions and simulation period, and the resultant low biases in simulated concentrations.*

Thanks for the comment. Here are responses to three main concerns and minor comments from the reviewer.

(1)  Fire and dust emissions

Episodic natural emissions from Asian dusts and wildfires have been seriously considered from the designing stage of the study. Technically, current modeling framework is capable of providing Asian dust and wildfire emissions, and we have studied several cases of fire and dust events (Bae et al. 2016). However, after initial assessment, we decided not to include modeling of Asian dust and wildfires in current regional emission attribution assessment. Estimated impact of dust and fire emissions in South Korea during 2014 was relatively small (<5%) and modeling of dust and fire emissions still have high uncertainties, especially in the threshold parameterization of friction velocity, for the dust emissions, and the magnitude and plume rise parametrizations, for the fire emissions.

Impacts of dust and fire emissions were investigated by filtering days with chances of dust or fire impact. For dust days, dust observations from the Korean Meteorological Administration (KMA) were used (http://www.kma.go.kr/weather/asiandust/observday.jsp). In 2014, 10 dust days were reported in the SMA. Discarding those days resulted in lowering the annual average of $PM_{10}$ concentration over the SMA by 3.2% (51.6 $\mu g/m^3$ to 49.9 $\mu g/m^3$). For fire days, there is no official report of fire impact observations. We utilized chemical component measurements (e.g. PM, OC, EC, K and SO4) at the Bulkwang supersite (at Seoul, lon=126.9, lat=37.6 ), satellite wildfire emissions data (QFED, based on MODIS fire radiative power, https://gmao.gsfc.nasa.gov/research/science_snapshots/global_fire_emissions.php), and NOAA HYSPLIT backward trajectory simulations. Considering probable fire emission chemical signals (high OC, EC and K, OC/EC ratio, and low $SO_4$) and daily backward trajectory pathways together, we selected 8 days of fire impact, including July 27-28 episodes as reported in Jung et al. (2016). Impact of those days were 0.4% (51.6 $\mu g/m^3$ to 51.4 $\mu g/m^3$). Figure R1 summarizes time series of surface $PM_{10}$ concentration over the SMA region in 2014. Days with (likely) dust and fire impacts are marked with "D" and "F". Examples of surface measurements and trajectory runs are shown in Figure R2 and Figure R3.

Still being preliminary, these calculations suggest that the estimated dust and fire impact in South Korea during 2014 might be smaller than other uncertainties we have discussed in current study. We

have clarified the use and (estimated) impact of dust and fire emissions in the manuscript. This analysis also will be  prepared as a separate research article later.

Bae, C. H., H. C. Kim, B.-U. Kim, and S. Kim, 2016: Implementation of Near Real-Time Fire and Dust Emissions in Air Quality Forecast over Northeast Asia, *17th IUAPPA Conference*, Busan, South Korea

Jung et al., 2016:  Impact of Siberian forest fires on the atmosphere over the Korean Peninsula during summer 2014, ACP, doi:10.5194/acp-16-6757-2016

(2)  Date of emission inventory

Selection of emissions inventory brings another big dilemma because a decision should be made between "reliable but old emission inventory" and "new but not fully tested emissions inventory". Current study was conducted for the year of 2014 because the UM model simulations (provided by KMA) are only available for the period. We are trying to extend current regional emission attribution estimation for long-term period, and preliminary results suggest no significant inter-annual variations (Figure R*4*).

(3)  Model bias

We agree that lack of Asian dust and wildfire emissions is one of the reasons for model low bias. However, we are hesitant to conclude that they can explain all the model biases. Simply, model biases happen regardless of dust or fire events, and current model bias is larger than our estimation of dust and fire impact. Currently, we have three hypothesis of model underestimation of PM concentration.
With current modeling system, modeling biases often occur by following reasons: (a) Uncertainty in emission inventory: Missing or old-dated emission emissions inventory. As the reviewer commented, lack of dust and fire emissions could explain a portion of model bias. (b) Missing chemical mechanisms: Unknown mechanisms of secondary organic aerosols formation or heterogeneous reactions (e.g. such as Sulfate formation on the surface of Asian Dust) (Baker et al., 2016; He et al., 2014; Xue et al., 2016). (c) Model wind overestimation. This issue is already mentioned in the current manuscript -- meteorological models sometimes fail to reproduce low wind speed (e.g. the stagnant condition (Ngan et al., 2013), resulting in the underestimation of simulated concentration. Based on discussion with meteorological modeler (S. Hong, personal communication), we may improve surface wind speed by adjusting background diffusivity, but he did not recommend it because it can ruin the predicted precipitation amount. We expect, wind bias issue will be improved as meteorological models develops.

The bottom line is that current model is not perfect, and may be limited in generating absolute amount of particulate matter concentration, but performs pretty well to simulate spatial and temporal variations. In terms of relative attribution assessment, we do not see any serious limitation in the current modeling system we have employed.

Baker, K.R., Woody, M.C., Tonnesen, G.S., Hutzell, W., Pye, H.O.T., Beaver, M.R., Pouliot, G., Pierce, T., 2016. Contribution of regional-scale fire events to ozone and PM2.5 air quality estimated by photochemical modeling approaches. Atmospheric Environment 140, 539–554. doi:10.1016/j.atmosenv.2016.06.032

He, H., Wang, Y., Ma, Q., Ma, J., Chu, B., Ji, D., Tang, G., Liu, C., Zhang, H., Hao, J., 2014.  Mineral dust and NOx promote the conversion of SO2 to sulfate in heavy pollution days. Scientific Reports 4. doi:10.1038/srep04172

Xue, J., Yuan, Z., Griffith, S.M., Yu, X., Lau, A.K.H., Yu, J. Z., 2016. Sulfate Formation Enhanced by a Cocktail of High NOx, SO2, Particulate Matter, and Droplet pH during  Haze-Fog Events in Megacities in China: An Observation-Based Mode ling Investigation.  Environ. Sci. Technol. 50, 7325–7334. doi:10.1021/acs.est.6b00768

Ngan, F., H. Kim, P. Lee, K. Al-Wali, B. Dornblaser, 2013, A study on nocturnal surface wind speed overprediction by the WRF-ARW model in Southeastern Texas, *J. of App. Meteo. and Clim.,* doi:10.1175/JAMC-D-13-060.1

*A second set of simulations to assess the role of transported biogenic emissions would be valuable, but less critical.*

Thanks for the suggestion. We will pursue this idea in our next study.

*Technical comment*
*The authors do not define the simulation period until the results section. Specific start and end dates for the simulation and any initialization period should appear in the first paragraph of section 2.*

Simulation period and spin-up time were included in section 2.

[Figure]

Figure R1 Time series of $PM_{10}$ concentration from surface monitoring sites (112 sites) in the SMA, Korea. 10-m wind speed and surface pressure are also shown in blue and red lines. Dust and fire cases are marked with "D" and "F", respectively.

[Figure]

Figure R2 Time series of PM$_{10}$, PM$_{2.5}$, OC, EC, K and SO$_4$ measured at the Bulkwang super site in Seoul, Korea.

[Figure]

Figure R3 7-day HYSPLIT backward trajectory simulations overlaid with QFED fire emission (CO emission during July 21-27), arriving at the Bulkwang super site on July 27, 2014, 12:00PM local time.

[Figure]

Figure R4 Inter-annual variation of estimated contributions from foreign emission sources (pink) and domestic source (blue). INTEX-B 2006 and CAPSS 2007 emissions inventories were used for international and South Korea, respectively.

---

## Author Comment (AC2) · 7 May 2017

**Authors' response to the review comments #1**

**"Regional Contributions to Particulate Matter Concentration in the Seoul Metropolitan Area: Seasonal Variation and Sensitivity to Meteorology and Emissions Inventory" by Kim et al.**

**General response**

The authors express their appreciation to the two reviewers and the editor. We believe that their comments are very productive and substantially contributed to improving the manuscript. We offer general responses and point-by-point responses to the issues and comments addressed by the reviewers. Reviewers' comments are shown in italics.

*Anonymous Referee #1*

*1. The findings are framed as being robust and representing uncertainty. However, this limited number of cases does not constitute a comprehensive ensemble or represent the range of uncertainty that may exist in the emissions inventory. The modeling relies on a somewhat arbitrary set of 4 emissions inventories, which differ for domestic and foreign sources and which do not directly represent the year (2014) that is simulated. Thus, it is not justified to call the findings robust (p. 1, line 25 and p. 12, line 19) and more caution is needed in interpreting the findings.*

Thanks for the comment. We understand the limitation of current study, and clarified the meaning of our findings in the manuscript. Our study intends to provide information of uncertainties of regional emission attribution depending on the selection of meteorology model and/or emissions inventories. We replaced the "robust" term with "mostly consistent".

*"We also found that simulated surface PM concentration is sensitive to meteorology, but estimated contributions are mostly consistent." "While not a comprehensive ensemble, simulations using multiple combinations of emissions inventories all showed similar seasonal variation."*

Selection of emissions inventory for regional air quality modeling is always an issue because we need to make a decision between "reliable but old emissions inventory" and "new but not fully tested emissions inventory". Current study is limited to the year 2014 due to the data availability. We are trying to extend this attribution estimation for longer period. Preliminary results during 2004-2015 show estimated contributions are mostly invariant year-by-year (Figure R*1*).

*2. Source contributions are defined by doubling the impact of 50% emission reduction runs. However, if the reductions are applied only to anthropogenic emissions (this was unclear), then some of what is being termed "foreign" is actually resulting from biogenic emissions within Korea. Also, zero-out impacts are often larger than 2x the impact of 50% out cases, due to nonlinearities of the chemistry of pollutant formation. If that is the case here, it would systematically under-represent the domestic contribution, and hence over-represent the foreign share. One run should be conducted to test the linearity of response from 50% to 100% reduction, and language should be more cautious in defining source apportionment if it is based on 50% cases.*

Sorry for the confusion. Emission reduction was applied to the total national emission (i.e. total biogenic and anthropogenic emission within South Korea). We have clarified it in the methodology section.

*"Here, we used a 50 % reduction in the South Korean national emission (e.g. total biogenic and anthropogenic emissions within South Korea) as a test."*

We also conducted an additional full-year simulation to test the linearity of the brute force method (BFM) of 50% domestic emission reduction. For the BFM method sensitivity test, we have compared two BFM runs using 50% reductions of domestic and foreign emissions. We think the concept of zero-out contribution could be ambiguous in the national emission level. Unlike local emission sources which can be practically zeroed-out, removing total national emissions is usually an unrealistic scenario. In an alternative approach, we tried to compare two 50%- emissions reduction methods for domestic emissions and foreign emissions. Seasonal variations of two BFM methods over the Seoul Metropolitan Area (SMA) and South Korea (SKR) are compared in Figure R2. As expected, responses to the reductions of domestic and foreign emission are not identical, showing non-linearity of responses. While estimated foreign emission contributions using different BFM methods show similar seasonal variations, high in winter and low in summer, month-to-month variations show a certain uncertainty range, ~10%, which is not much different from the uncertainties from meteorology model selection or emissions inventory selection.

**3. The model substantially under-estimates observed PM. This raises serious doubt about the conclusions, since it could indicate error in either the domestic or foreign emissions inventory.**

Thanks for the comment. We agree that missing emission sources (e.g. dust and fire emissions and/or other unknown emission sources) might be associated with current model bias, but do not think that current model bias raises any critical issue in the quality of emission inventories. Currently, we suspect there are several issues that can cause model low biases. In current modeling system, modeling biases often occur by following reasons: (a) Uncertainty in emission inventory: Missing or old-dated emission emissions inventory. As the reviewer commented, lack of dust and fire emissions could explain a portion of model bias. (b) Missing chemical mechanisms: Unknown mechanisms of secondary organic aerosols formation or heterogeneous reactions (e.g. such as Sulfate formation on the surface of Asian Dust) (Baker et al., 2016; He et al., 2014; Xue et al., 2016). (c) Model wind overestimation. This issue is already mentioned in the current manuscript -- meteorological models sometimes fail to reproduce low wind speed (e.g. the stagnant condition) (Ngan et al., 2013), resulting in the underestimation of simulated particle concentration. Based on discussion with meteorological modeler (S. Hong, personal communication), we may improve the surface wind speed by adjusting background diffusivity, but he did not recommend it because it can ruin the predicted precipitation amount. We expect, wind bias issue will be improved as meteorological models further develop.
The bottom line is that current model is not perfect, and may be limited in generating absolute amount of particulate matter concentration, but performs pretty well to simulate spatial and temporal variations. In terms of relative attribution assessment, we do not see any serious limitation in the current modeling system we have employed.

Baker, K.R., Woody, M.C., Tonnesen, G.S., Hutzell, W., Pye, H.O.T., Beaver, M.R., Pouliot, G., Pierce, T., 2016. Contribution of regional-scale fire events to ozone and PM2.5 air quality estimated by photochemical modeling approaches. Atmospheric Environment 140, 539–554. doi:10.1016/j.atmosenv.2016.06.032

He, H., Wang, Y., Ma, Q., Ma, J., Chu, B., Ji, D., Tang, G., Liu, C., Zhang, H., Hao, J., 2014.  Mineral dust and NOx promote the conversion of SO2 to sulfate in heavy pollution days. Scientific Reports 4. doi:10.1038/srep04172

Xue, J., Yuan, Z., Griffith, S.M., Yu, X., Lau, A.K.H., Yu, J. Z., 2016. Sulfate Formation Enhanced by a Cocktail of High NOx, SO2, Particulate Matter, and Droplet pH during Haze-Fog Events in Megacities in China: An Observation-Based Mode ling Investigation. Environ. Sci. Technol. 50, 7325–7334. doi:10.1021/acs.est.6b00768

Ngan, F., H. Kim, P. Lee, K. Al-Wali, B. Dornblaser, 2013, A study on nocturnal surface wind speed overprediction by the WRF-ARW model in Southeastern Texas, *J. of App. Meteo. and Clim.,* doi:10.1175/JAMC-D-13-060.1

*Minor suggestions: p. 2, line 28: "region's"*
Corrected

*p. 2, line 33: excess precision in numbers*
Corrected

*p. 3, lines 3-6: Meteorological uncertainty has been studied elsewhere, with larger ensembles than considered here*

Thanks for the comment. We know that meteorological ensembles have been studied actively, but those uncertainties in emission attribution in South Korea are not well addressed. We have clarified our point in the manuscript.

*"While many studies have addressed sources of uncertainties in the estimation of contributions or source apportionment, few have tried to investigate, whether qualitatively or quantitatively, the uncertainties resulting from the meteorological model, especially in the estimation of source contributions in South Korea."*

*p. 6, line 26: "compromising" is the wrong word*

We have modified the sentence.

*"However, the BFM still provides efficient and practical way of analyzing source contributions."*

*p. 9, lines 1-7: These explanations are not convincing, and the discussion of specific days is not helpful*

Thanks for the comment. As discussed in the model bias comment, we do not have clear evidence to specify the reason of model low bias from multiple possibilities. Manuscript was modified to mention this point.

*"One typical problem in the chemical modeling of surface PM in the SMA is that simulated surface PM concentration constantly underestimates observed measurements. Low bias can happen with several reasons: missing or old-dated emission sources, lack of dust and fire emissions, unknown chemical mechanisms, and meteorological bias, such as the wind bias discussed in this study. At this point, we do not have clear evidence to specify the reason. In general, surface PM concentration simulated using UM-CMAQ generates higher PM concentrations compared to the WRF-CMAQ system, which we suspect results from UM-CMAQ's weaker wind field, which results in a more stagnant and shallower boundary layer."*

*p. 12, line 25: How can results be "considerable but not significant"?*

We replaced the sentence.

*"We found that differences in meteorological model can lead to discernible differences in the estimation of contributions from regional (e.g., domestic and international) emissions sources although they still have similar seasonal patterns."*

[Figure]

Figure R1 Inter-annual variation of estimated contributions from foreign emission sources (pink) and South Korean domestic source (blue). INTEX-B 2006 and CAPSS 2007 emissions inventories were used for international and South Korean emissions inventories, respectively.

[Figure]

*Figure R2 Estimated contributions from foreign emission sources in 2014 over the Seoul Metropolitan Area (SMA) and over South Korea (SKR). Two BFM methods, reducing 50% of domestic emissions or foreign emissions, were compared.*